# Computer Vision System for Welding Inspection of Liquefied Petroleum Gas Pressure Vessels Based on Combined Digital Image Processing and Deep Learning Techniques

**DOI:** 10.3390/s20164505

**Published:** 2020-08-12

**Authors:** Yarens J. Cruz, Marcelino Rivas, Ramón Quiza, Gerardo Beruvides, Rodolfo E. Haber

**Affiliations:** 1Centro de Estudio de Fabricación Avanzada y Sostenible, Universidad de Matanzas, Matanzas 40100, Cuba; marcelino.rivas@umcc.cu (M.R.); ramon.quiza@umcc.cu (R.Q.); 2Social Innovation Business, Hitachi Europe Ltd., 40547 Hitachi, Germany; gerardo.beruvides@hitachi-eu.com; 3Centro de Automática y Robótica, CSIC-UPM, 28500 Madrid, Spain; rodolfo.haber@car.upm-csic.es

**Keywords:** computer vision, digital image processing, deep learning, welding inspection

## Abstract

One of the most important operations during the manufacturing process of a pressure vessel is welding. The result of this operation has a great impact on the vessel integrity; thus, welding inspection procedures must detect defects that could lead to an accident. This paper introduces a computer vision system based on structured light for welding inspection of liquefied petroleum gas (LPG) pressure vessels by using combined digital image processing and deep learning techniques. The inspection procedure applied prior to the welding operation was based on a convolutional neural network (CNN), and it correctly detected the misalignment of the parts to be welded in 97.7% of the cases during the method testing. The post-welding inspection procedure was based on a laser triangulation method, and it estimated the weld bead height and width, with average relative errors of 2.7% and 3.4%, respectively, during the method testing. This post-welding inspection procedure allows us to detect geometrical nonconformities that compromise the weld bead integrity. By using this system, the quality index of the process was improved from 95.0% to 99.5% during practical validation in an industrial environment, demonstrating its robustness.

## 1. Introduction

Pressure vessels are closed containers, tanks, or pipelines designed to receive and store a fluid at a pressure greater than outer ambient pressure conditions. They are used in a large number of industries, such as power generation, chemical, petroleum, petrochemical, and nuclear industries. The fluids contained in pressure vessels may have specific characteristics, such as volatility, compressibility, flammability, or radioactivity. Cylindrical vessels are generally preferred, because they present simpler manufacturing problems and make better use of the available space [1].

Modern industries need to reduce the production time in order to accomplish their customers’ demands, and to achieve higher profits; industries that produce pressure vessels are not the exception. As a consequence, it is still a challenge to apply systematic methodologies for monitoring and preventing defect occurrences within the manufacturing shop floors, due to the increasing complexity of both products and production systems [2]. If the defects originating during the manufacturing process are not detected, the vessel can break during operation. A ruptured pressure vessel can be hazardous, possibly leading to poison gas leaks, fires, or explosions, which may cause significant losses of human lives and properties [3]. Thus, the application of more efficient monitoring strategies on the production lines at critical stages is required to avoid the generation or propagation of defects [4].

Various methodologies can be adopted to manufacture a vessel. Nonetheless, the basic manufacturing process of a pressure vessel consists of the following stages: forming, pressing, spinning, bending, welding, post-weld heat treatment, assembly, and finishing. During the manufacturing of liquefied petroleum gas (LPG) pressure vessels, a large number of defects originate in the welding stage. The use of non-destructive testing (NDT) techniques can contribute to the early detection of these defects, allowing the deployment of cost-effective line monitoring and control systems that reduce expensive off-line measure-rework-assess loops. NDT techniques, such as radiography, ultrasonic testing, penetrant liquid testing, magnetic particle testing, phased arrays, time-of-flight diffraction, and multi-elements eddy current, are more and more extensively applied. Tomography, acoustic emissions, ultrasonic guided waves, and laser ultrasonic techniques continue to be strong topics of interest [5].

Lately, NDT techniques based on computer vision are becoming integral parts of many production systems, due to the increasing computing power, hyper connectivity, and easy installation of digital cameras in production lines. Many 3-D light detection and ranging (LiDAR) techniques have been explored and reported in the literature [6,7]; however, these sensing systems are expensive and require a high computational cost. Nowadays, computer vision technologies have demonstrated unprecedent benefits in the industry; they allow detection of defects unnoticeable to human operators, automate extremely tedious measuring tasks, perform visual inspection in risky environments, substitute costly end-of-line product quality inspection procedures for multi-stage inspection systems, etc.

A wall thickness of 2 mm is typically used for 10 kg LPG pressure vessels. Due to the thinness of the wall, it is critical to achieve a high precision in the butt joint alignment before welding. The use of structured light vision is widely spread for alignment evaluation, since it allows reduction of the effect of visual perturbations commonly found on shop floors.

A multi-functional monocular visual sensor, based on the combination of cross-lines and single-line laser structured lights, was proposed by Jichang et al. [8]. This method allowed determination of the dimensions of V-type butt groove based on a single image processing. Wang et al. [9] developed a robust weld seam recognition method under heavy noise based on structured light vision. By using this algorithm, butt joints, T joints, and lap joints were accurately recognized. Shao et al. [10] used combined passive and active vision sensors to measure dimensions in butt joints with seam gap less than 0.1 mm. Later, Shao et al. [11], continued to develop the previously mentioned system by using a particle filter in order to make it more robust. Fan et al. [12] introduced a method based on digital image processing for initial point alignment in narrow robotic welding. This method allowed detection of the weld seam center point when a laser stripe line was projected over a junction, and it was used later by Fan et al. [13] to develop a weld seam tracking system. Robertson et al. [14] used a Keyence blue laser profilometer to digitize the geometry of the weld seam; this permitted them to formulate an automated welding plan. Recently, Chen et al. [15] introduced a method for the inspection of complex joints, which allowed them to assign different parameters for the welding process, depending on the features of the region. Du et al. [16] proposed a convolutional neural network (CNN) to perform the feature area recognition and weld seam identification. This CNN analyzed the projection of a 659 nm laser stripe over different types of junctions, obtaining a validation accuracy of 98.0% under strong noise.

Welding defects and nonconformities are not only caused by pre-welding conditions. Some of these defects are intrinsic to the welding operation itself. For this reason, it is necessary to assess the weld bead integrity, especially in the case of pressure vessels, where the weld bead is going to be under constant stress. The use of structured light vision is the core of many studies, while other computer vision approaches propose the analysis of X-ray images, infrared images, ultraviolet images, etc.

Pinto-Lopera et al. [17] proposed a system for measuring weld bead geometry by using a single high-speed camera and a long-pass optical filter. Soares et al. [18] introduced a method for weld bead edge identification that allowed them to recognize discontinuities. Han et al. [19] and Zhou et al. [20] used the RANSAC algorithm for fitting linear functions to a laser stripe over a welded surface; which allowed them to measure weld bead dimensions. Ye et al. [21] investigated the use of a model-based classification method to automatically segment the bead from the welding surface, regardless of the distance and the angle of the scanner to the welding surface. An approach based on pixel intensity analysis was proposed by Singh et al. [22] for studying weld beads in P-91 steel plates. Leo et al. [23] introduced a system for detecting and following the internal weld bead in stainless steel kegs. Zeng et al. [24] proposed a weld bead detection method based on light and shadow feature construction, using directional lights. Dung et al. [25] compared three deep learning methods for crack detection in gusset plate welded joints. Khumaidi et al. [26] and Zhang et al. [27] used CNN for weld inspection, reaching accuracies of 95.8% and 93.9%, respectively, for the classification of three different types of defects, while the CNN developed by Bacioiu et al. [28] achieved a 93.4% accuracy during the classification of five different types of defects. The transfer learning approach was used by Yang et al. [29] for optical inspection of laser welding. A deep neural network model that extracted the intrinsic features of X-ray images was used by Hou et al. [30] for automatic weld defect detection, reaching an accuracy of 97.2%.

Although several researches on pre-welding and post-welding inspection exist, they are usually only focused on one of these tasks. There is a lack of studies on the combined use of computer vision techniques to provide an integral welding inspection under practical shop floor conditions, which may include variation in illumination, presence of fume, and mechanical vibrations. For dealing with the aforementioned shortcomings, this paper proposes an integrated system for pre-welding and post-welding inspection, based on computer vision techniques, for industrial applications.

## 2. Materials and Methods

In the welding stage of LPG pressure vessels analyzed, most defects were related to butt joint misalignment and incorrect weld bead geometry. For this reason, the proposed system should be able to detect both types of imperfections. The welding inspection system proposed in this paper includes a 650 nm line laser projector with a fan angle of 90°, a Raspberry Pi (RPi) 3B, an 8 MP Raspberry Pi Camera Module V2, an Omrom E6B2 rotary encoder, a display, and a server. The sensors were installed in the welding stage of the LPG pressure vessel production process. In this stage, the vessels were mounted in an automated device that allowed rotation of them. The laser line was projected orthogonally to the joint, with an incidence angle of 30° over the vessel surface from a distance of 500 mm. The camera was conveniently placed to capture the laser incidence area from a distance of 72 mm. The area captured by the camera was 65 × 87 mm. Rotating the vessel allowed the capture of images along the circumferential junction before and after the welding operation. A rotational offset of 6° between images is proposed. The rotary encoder was used in order to determine the precise moment for capturing the images. All the processing of the data was carried out in the RPi. Relevant information was shown in the display and also stored in the server for further analysis. Figure 1 shows an overview of the system, while Figure 2 shows images captured before and after the welding operation. The images in Figure 2 were cropped vertically for displaying the area of interest, however, the horizontal size remained the same, corresponding to 65 mm on the vessel surface.

In the production line, the inspection system was implemented in two separate steps: a pre-welding inspection for detecting misalignment in the parts to be joined, and a post-welding inspection directed to estimate the weld bead dimensions and to verify if they accomplished the technical requirements. Figure 3 presents the block diagrams for both inspection tasks. In the following subsections, the steps of these algorithms are explained.

### 2.1. Pre-Processing

The pre-processing step was the same for both inspection tasks. After being captured, images were, firstly, resized to 640 × 480 pixels, and then cropped to 320 × 480 pixels. Next, they were converted from RGB color space to grayscale, because information about hue and saturation is not significant for processing. A third transformation was conducted to separate the laser line from the background. A binarization based on an adaptive threshold allowed us to reach this objective. The threshold was calculated using the method proposed by Bradley and Roth [31], based on the local mean intensity in the neighborhood of each pixel. The resulting images have just one value per pixel: 0 or 1; indicating black or white, respectively. After this transformation, only the biggest object of each image was preserved, this contributed to eliminating smaller objects produced by ambient noise. In order to simplify the analysis, the laser line was reduced to a one-pixel width line. An example demonstrating the previously described transformations can be seen in Figure 4. After these operations, images were ready to be processed.

### 2.2. Pre-Welding Image Processing and Decision Making

In the last decade, artificial intelligence has been widely applied in pattern recognition. Among available techniques, image classification using CNN has been reported in many studies [32,33,34,35]. This method has demonstrated to learn interpretable and powerful image features after the correct training. For this reason, it was proposed as the processing method for evaluating the pre-welding images that needed to be classified into two categories: “correct alignment” and “incorrect alignment”. Due to the characteristics of the images, a CNN architecture containing few layers was proposed. This architecture is shown in Figure 5.

The purpose of the first layer is to serve as an input to the network by mapping each pixel of every image received into a matrix. The 2-D convolutional layer performs a convolution operation on the output of the preceding layer, using a set of filters in order to learn the features relevant to distinguishing a “correct alignment” image from an “incorrect alignment” image. The convolution operation is described by the following equation:(1)$$G\left[m,\;n\right]=\left(f*h\right)\left[m,\;n\right]=\sum_{j}\sum_{k}h\left[j,\;k\right]f\left[m-j,\;n-k\right],$$
where *f* is the input image mapped into a matrix, *h* is a filter, and *m* and *n* are the rows and columns, respectively. In this case, a total of 64 filters were used, and the activation function was “relu”, described by the following equation:(2)f(x)=max(0, x),
which is commonly used in deep neural networks.

The pooling layer downsamples the results of the preceding layer by discarding features, which helps make the model more general, and reduces computation time. The pooling technique used was “max pooling”, which examines squares of 2 × 2 features, and keeps only the maximum value. Dropout layers randomly remove certain features by setting them to zero in each training epoch, this is a commonly used technique to prevent overfitting. The flatten layer allows reshaping of the three-dimensional data received from the first dropout layer into one dimension. Dense layers or fully-connected layers learn the relationships among features, and perform classification by connecting every neuron in the preceding layer to all the neurons they contain. The first dense layer contained 128 neurons and their activation function was “relu”, while the second dense layer contained only 2 neurons, corresponding to the number of categories that needed to be classified, and their activation function was “softmax”, described by the following equation:(3)$$\sigma {\left(z\right)}_{i}=\frac{e^{z_{i}}}{\sum_{j=1}^{K}e^{z_{j}}},\;\mathrm{for}\;i=1,\;\ldots ,\;K,$$
where *z* is, in this case, the output of the preceding layer. This function allows us to interpret the output of the network as probabilities, where the category with the highest value will be the predicted class.

For pre-welding inspection, a total of 60 images per vessel are taken and analyzed one at a time. The CNN classifies each image as “aligned” or “misaligned”. When the first misaligned image is detected, the inspection process is interrupted, the vessel is labeled as defective, and it is removed from the production line. If all the images are classified as aligned, the vessel is labeled as acceptable, and is submitted to the welding operation. All the information generated by the system is stored in the server.

### 2.3. Post-Welding Image Processing and Decision Making

CNN capabilities for image classification are remarkable. However, the processing of post-welding images implies a different task, to accurately estimate the weld bead dimensions. For this reason, another approach based on the examination of the laser profile was proposed. In order to determine where the weld bead is located in the image, the first and second derivative of the one-pixel width laser profile are calculated using the numeric approximations described in the following equations, respectively:(4)$$f^{\prime }\left(x_{0}\right)\;\approx \;\frac{f\left(x_{0}+h\right)-f\left(x_{0}-h\right)}{2h},$$
(5)$$f^{\prime \prime }\left(x_{0}\right)\;\approx \;\frac{f\left(x_{0}+h\right)-2f\left(x_{0}\right)+f\left(x_{0}-h\right)}{h^{2}},$$
where *h* is a step value.

The minimum value of the second derivative proved to be a representative feature of the weld bead center. The maximum values of the second derivative on both sides of the weld bead center proved to be representative features of the weld bead edges. Figure 6 shows the mentioned points in the second derivative graphic representation, while Figure 7 shows the same points over the RGB image.

The estimations of the weld bead dimensions are performed using a laser triangulation algorithm. Calculating the distance between the weld bead edges feature points (left and right features points of Figure 7) allows estimation of the weld bead width, *W*, in pixels. Calculating the distance between the weld bead center feature point and the line joining the weld bead edges feature points allows estimation of the weld bead height, *H*, in pixels. These values are converted to millimeters by applying a scale factor depending on the distance between the camera and the examined area.

For post-welding inspection, 60 images per vessel are also analyzed, one at a time. For each of them, the weld bead height and width are computed. If, for all the images, these values are in the corresponding intervals (i.e., 0.5 mm ≤ *H* ≤ 1.5 mm and 7.0 mm ≤ *W* ≤ 9.0 mm), the vessel is labeled as acceptable, and is submitted to further operations. On the contrary, if, for a given image, any of these values are below the lower limit (i.e., *H* < 0.5 mm or *W* < 7.0 mm), the inspection process is interrupted, the vessel is classified as defective, and is removed from the production line. Finally, if, for a given image, any of the dimensions are above the upper limit of the corresponding intervals (i.e., *H* > 1.5 mm or *W* > 9.0 mm), the vessel is still labeled as acceptable, but a warning is emitted to the operator for considering a possible readjustment of the welding parameters, for preventing material and energy wasting. All the information generated by the system is stored in the server.

## 3. Results

A dataset containing 1090 images, divided into 872 images for training and 218 for validation, was used for the parametrization of the CNN. Of the total images, 565 had a misalignment value inferior to 1 mm and were labeled as “correct alignment”; and 525 images had a misalignment value superior to 1 mm and were labeled as “incorrect alignment”. During training, the accuracy of the CNN rapidly grew in the first epochs, and after the 35th epoch was superior to 95.0%, as shown in Figure 8. To prevent overfitting, the model was validated after each training epoch by evaluating it with unseen data. The validation loss reached its minimum value in the 44th epoch, as shown in Figure 9. After this epoch, there was no improvement of the validation loss during 20 consecutive epochs, and for this reason, the training was early stopped. The network parameters for the epoch where the best validation was achieved were recovered and set to the CNN.

A new dataset containing 109 “correct alignment” images and 111 “incorrect alignment” images was used for testing the model. Table 1 shows the results obtained during testing. The accuracy obtained was 97.7%, which demonstrates that the CNN was able to learn the relevant features for classifying the images with the occurrence of few errors, and it was concluded that this algorithm was satisfactory for the pre-welding inspection task. Additionally, the system was designed to store all the data in the server; this will allow the creation of a larger dataset in the future for retraining the model and obtaining a better accuracy.

With the objective of testing the accuracy of the post-welding inspection method, 10 vessels were analyzed. For each vessel, a total of 60 measurements were verified. The results are resumed in Table 2. The values correspond to the maximum absolute error (MAE) and maximum relative error (MRE) per variable.

Table 3 shows the comparison between the results obtained by the method used in the proposed system and others found in the literature, demonstrating that the errors produced during the estimation of the weld bead dimensions are within the common range for both variables. After this comparison, it was concluded that the proposed algorithm was satisfactory for the post-welding inspection task.

## 4. Discussion

In order to evaluate the performance of the proposed integrated system under actual shop floor conditions, the behavior of a batch containing 1000 vessels was analyzed. In every step, all the items were evaluated for determining if they conformed to the requirements of the corresponding operations. A comparison was carried out between the previously existing human-based inspection system and the proposed computer vision approach. Figure 10 shows the obtained results.

As can be seen, in the pre-welding inspection, the computer vision system showed a better performance, not only rejecting a higher number of nonconforming items (96.3%) than the human-based system (47.5%), but also wrongly rejecting a lower number of conforming items (0.4% vs. 1.6%). As the welding of misaligned parts produced dimensionally incorrect weld joints, the fraction of nonconforming items after the welding operation was also higher for the human-based inspection system (11.1% vs. 7.3%). Finally, in the post-welding inspection, the ratio of true positives was remarkably higher for the computer vision system (94.0% vs. 59.0%), while presenting a slightly lower false negatives ratio (1.6% vs. 3.3%).

From an overall analysis of the whole system, it can be noted that introducing the computer vision inspection increased the process quality index (i.e., the ratio between the conforming items and the total produced items) from 95.0% to 99.5%. It should be remarked that this is an important step toward a zero defects production. From the productivity point of view, the welding conforming index (i.e., the ratio between the final conforming items and the welded items) was notably higher for the computer vision system (91.2% vs. 86.0%). This last index is especially important, as the welding operation has a significant economic and environmental impact in the lifecycle assessment of the produced vessels.

## 5. Conclusions

As the main outcome of the paper, a computer vision inspection system, for detecting joint misalignment and geometrical defects in a welding process, was designed and implemented. In spite of its low cost, the used hardware was shown to be effective for achieving the proposed goal, and demonstrated a robust performance under the shop floor conditions where it was tested. In the pre-welding inspection, the used CNN was capable of detecting misalignment in 97.7% of the cases during the method testing. On the other hand, the laser triangulation approach used in the post-welding inspection estimated the weld bead dimensions with an average relative error of 3.4% for the weld bead width and 2.7% for the height during the method testing. The improvement of the overall quality index of the process during practical validation, from 95.0% to 99.5%, supported the technical feasibility of the industrial introduction of the proposed system.

As future development of the present work, it might be considered the incorporation of connectivity capabilities to the implemented modules, through the concepts of the Industrial Internet of Things. This addition will be an important step toward the integration of the considered welding process to a modern manufacturing environment with interconnected production stations and lines.

## Figures and Tables

**Figure 1 sensors-20-04505-f001:**
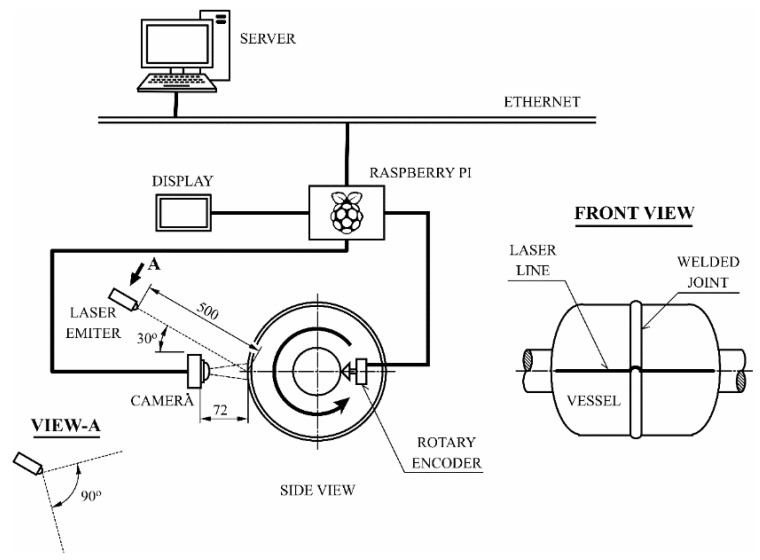
System overview.

**Figure 2 sensors-20-04505-f002:**
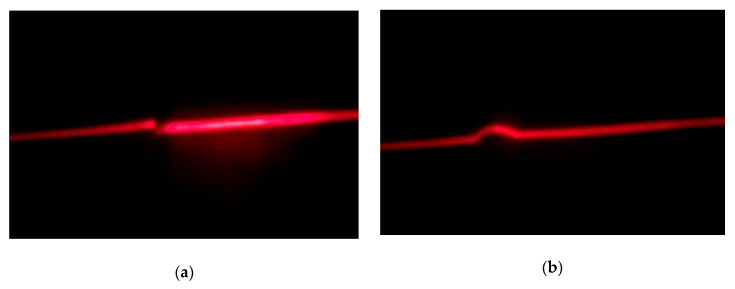
Images captured with the proposed system: (**a**) before welding operation; (**b**) after welding operation.

**Figure 3 sensors-20-04505-f003:**
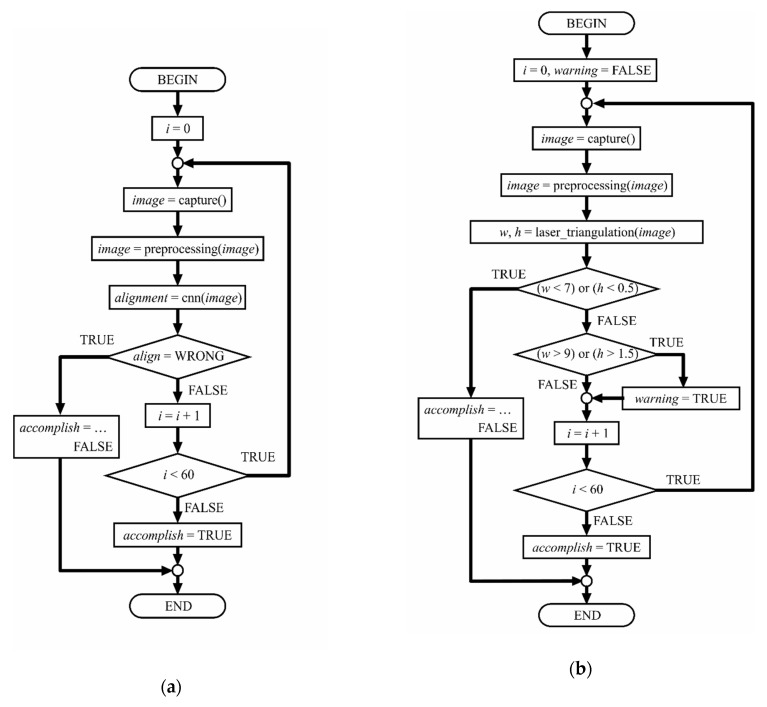
Block diagrams of the inspection tasks: (**a**) pre-welding inspection; (**b**) post-welding inspection.

**Figure 4 sensors-20-04505-f004:**
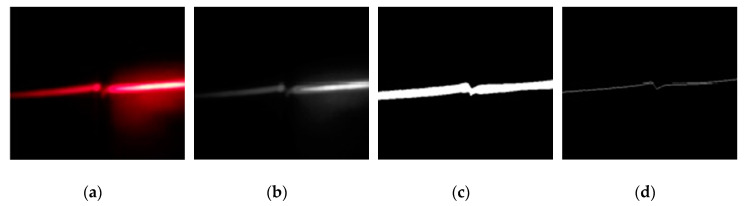
Images pre-processing: (**a**) RGB color space; (**b**) grayscale; (**c**) binarized image; (**d**) one-pixel width laser profile.

**Figure 5 sensors-20-04505-f005:**
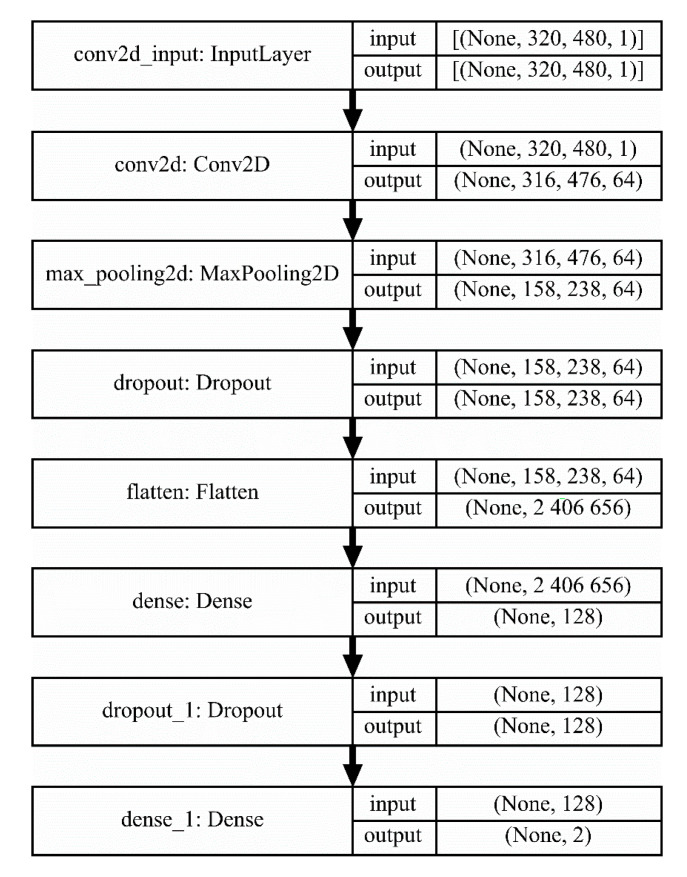
Convolutional neural network (CNN) architecture.

**Figure 6 sensors-20-04505-f006:**
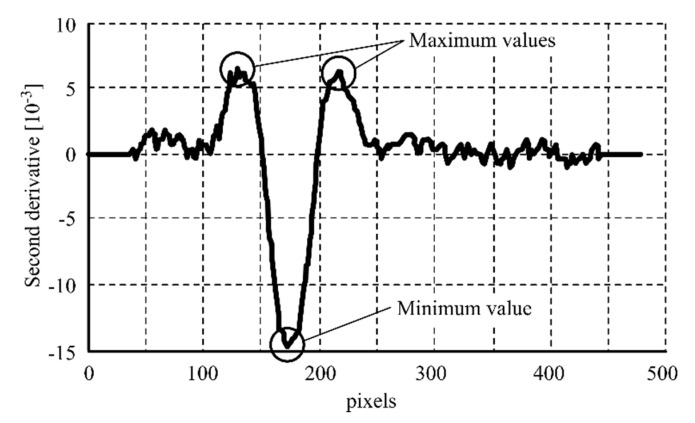
Second derivative of the one-pixel width laser profile after the welding operation.

**Figure 7 sensors-20-04505-f007:**
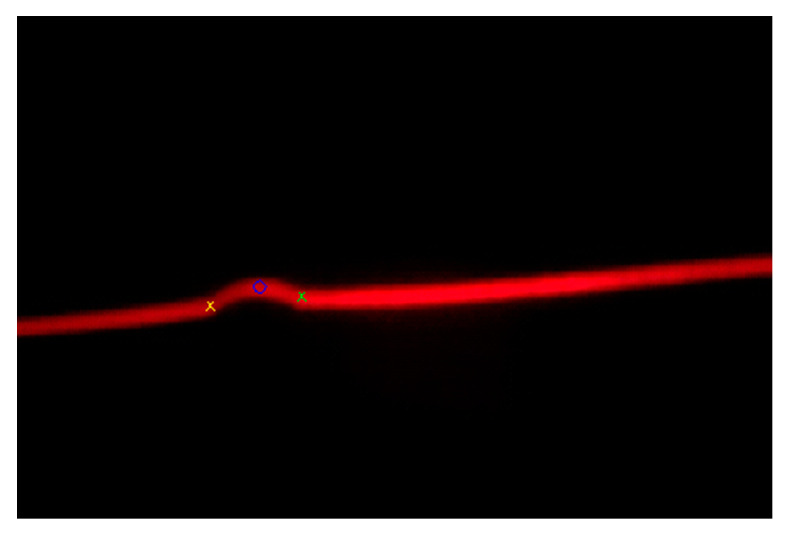
Feature points detected over the RGB image.

**Figure 8 sensors-20-04505-f008:**
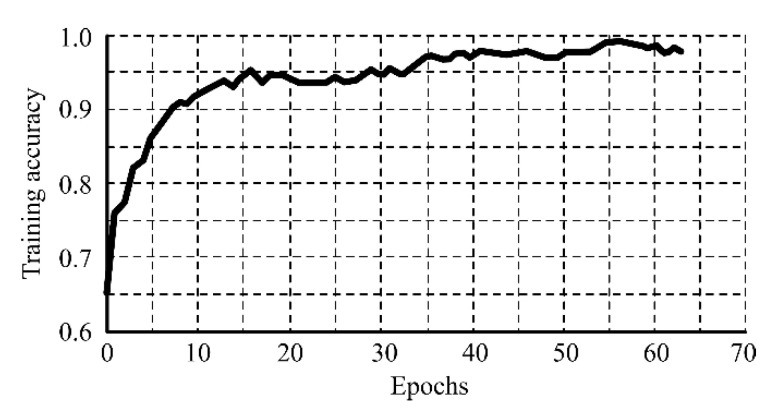
CNN accuracy during training.

**Figure 9 sensors-20-04505-f009:**
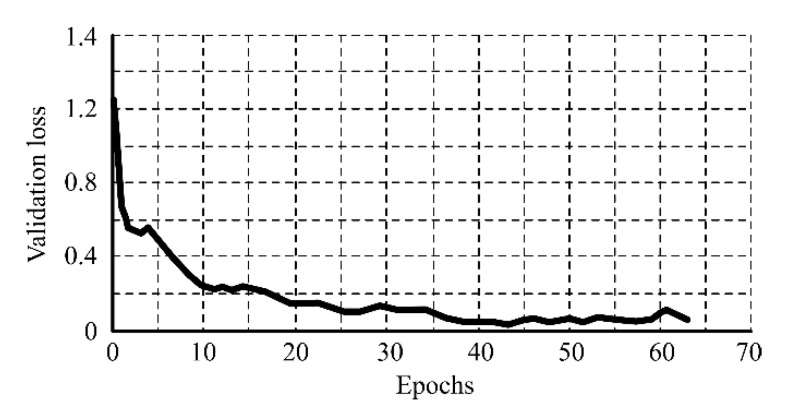
CNN validation loss for every training epoch.

**Figure 10 sensors-20-04505-f010:**
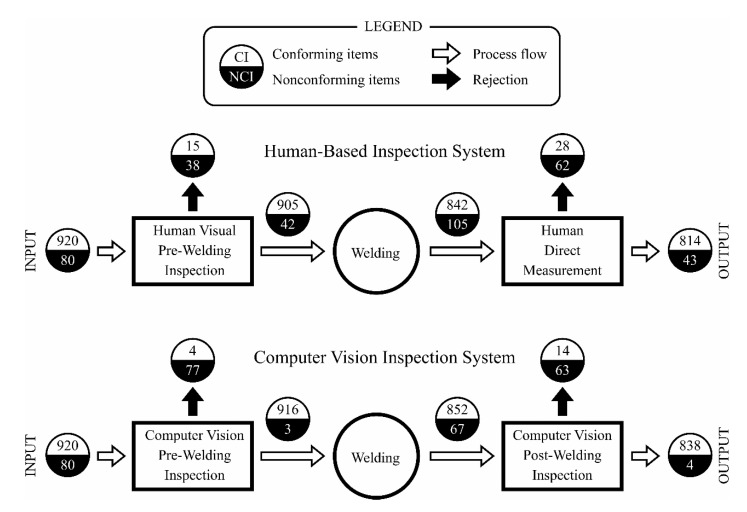
Practical validation results of the proposed system vs. human-based inspection.

**Table 1 sensors-20-04505-t001:** Confusion matrix for test dataset.

Total Images = 220	Actual “Correct Alignment”	Actual “Incorrect Alignment”
Predicted “correct alignment”	105	1
Predicted “incorrect alignment”	4	110

**Table 2 sensors-20-04505-t002:** Maximum absolute error and maximum relative error per variable.

Vessel	MAE Width	MRE Width	MAE Height	MRE Height
1	0.29 mm	3.9%	0.05 mm	3.5%
2	0.24 mm	3.4%	0.04 mm	2.8%
3	0.30 mm	4.2%	0.03 mm	3.0%
4	0.21 mm	2.5 %	0.03 mm	2.0 %
5	0.25 mm	3.8 %	0.03 mm	2.4 %
6	0.26 mm	4.1 %	0.04 mm	3.2 %
7	0.28 mm	3.9 %	0.02 mm	1.7 %
8	0.31 mm	4.2 %	0.03 mm	2.5 %
9	0.17 mm	2.0 %	0.03 mm	2.7 %
10	0.19 mm	2.5 %	0.05 mm	3.4 %
Average	0.25 mm	3.4 %	0.04 mm	2.7 %

**Table 3 sensors-20-04505-t003:** Methods comparison.

System	Average Absolute Error for Width	Average Absolute Error for Height
Method used in the proposed system	0.25 mm	0.04 mm
Method A [20]	0.25 mm	0.05 mm
Method B [36]	0.15 mm	0.11 mm

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
