# Peer review of "Computer Vision System for Welding Inspection of Liquefied Petroleum Gas Pressure Vessels Based on Combined Digital Image Processing and Deep Learning Techniques"

_sensors, 2020, doi:10.3390/s20164505_

Round 1

Reviewer 1 Report

1.This paper uses the CNN method to process the welding image, but it does not highlight the innovation. Please further highlight the process of using this method to illustrate, such as adding process block diagram, calculation process, etc.

2.The abstract should indicate the specific result, such as how much the accuracy has been improved

3."Conclusion" should be added to summarize the full paper

Author Response

We highly appreciate your comments and suggestions. Taking them into consideration we have made a revision of the paper, and the aspects pointed by you were addressed as follows:

  1. We have added block diagrams for the pre-welding and post-welding inspection procedures in Figure 3, where the use of the CNN in the pre-welding alignment classification is clarified. We have also added equations (1 - 3), related to the operations carried out by the network. The results of the network training, validation and testing have been moved to the Results section, where we have included Table 1 showing the confusion matrix of the network during testing. Discussion section have been rewritten to clarify the results of the proposed system during practical validation; Figure 10 have been added and now the improvements achieved by the system are more evident. Consequently, we have made changes in the Abstract and Conclusions sections to highlight the innovation of the system.
  2. As the reviewer suggests, the accuracy of the proposed system should be clearly addressed in the paper. In this sense, the Results and Discussion sections were completely rearranged and rewritten. A new approach was used for evaluating the system through the comparison with the performance of the previously existing human-based inspection, resumed in Figure 10. The obtained improvements were highlighted. Error values for dimension estimations were also included and compared with previously existing works. In the abstract, there were included the numerical values of the main performance indexes.
  3. We have included Conclusions section to summarize the full paper.

Reviewer 2 Report

The paper deals with the welding inpection by using computer vision based methods, but where does the novelty lie in this paper? The data processing approaches are common, and the experiments based on laser and camera are not new enough, as laser and camera measurements are widely used nowadays. 

The authors criticize the studies that validated using laboratory conditions, hodw did you know that these methods did not work for your case, as you did not present any comparison. 

Please improve the quality of Figure 5. 

Author Response

We highly appreciate your comments and suggestions. Taking them into consideration we have made a revision of the paper, and the aspects pointed by you were addressed as follows:

  1. This paper does not propose a single method for joint misalignment classification or measuring the weld joint dimensions, but an integral system which includes pre-welding and post-welding inspections, under practical workshop-floor conditions. To the best of our knowledge, no paper has been found based on this approach. Proper changes have been made for reflecting this conception through the whole paper, especially in the Abstract, Introduction, Discussion, and Conclusions sections.
  2. We agree we have not evidence on drawbacks of previous approaches tested under laboratory conditions for our specific case. We have reformulated the idea in lines 106-111. Although estimating the weld joint dimensions is not the unique goal of this work, we included a comparison of the obtained outcomes with those presented by previously reported studies. Table 3 was included for summarizing the comparison.
  3. We have improved the quality of Figure 5 (tiff 1000 dpi).

Reviewer 3 Report

I appreciate that the authors present a real industrial application, which is significant. however, several parts can be improved.

1-Throughoout all the paper, for example in line 17, in line 266, the authors should clearly state that their method detects only geometrical defects. they too often talk of defects, but welding defects are of manyother  types, cracks, porosity, ...... and the method presented cannot detect them.

2-Several references are questionable.

  • for example, ref. [1] is used to define what is a pressure vessel. this paper is dated 2019, while pressure vesseles exist since more than 100 years ! Why do they choose this extremely recent reference. Either they explain it, or they delete it.
  • again, why they use reference [3] to support a sentence that says "industries that produce pressure vessels are not the exception ?" I don't see any evident link between the paper referenced and the content of the sentence.
  • again, reference [7]. Why should it be related to "more efficenit monitoring strategies " ? Either explain, or delete the reference.

3-In line 34 I read "it is much more challengeing to apply ...."; my question is: much more challenging than what ? Please rephrase.

4-Line 49 - use "penetrant liquid testing"; in the text liquid is missing.

5-line 49. the acronym TOFD is not defined in the paper.

6-In line 55 you should write 3-d LiDAR techniques, instead of 3-D laser scanning. In fact, references [9, 10] are about LiDAR.

7-The description of the optical lay-out need a substantial improvement:

  • describe the relative position of the laser line with respect to the weld; I guess from the pictures that laser line is orthogonal to the weld, but this needs to written.
  • in figure 1 the laser is not represented as a laser line projector. If the drawing is correct, then the laser line would be parallel to the weld, which I do not understand. Please improve the picture by showing the laser sheet with its fan angle.
  • quantitative information on the laser traingulation system should be added: at least the angle between the optical axis of the camera and the optical axis of the laser projector, and the fan angle of the lasere line projector and the stand-off distance of the laser triangulation system with respect to the weld.
  • The dimension of the inspected area has to be declared. How long in the laser line imaged in figs. 2, 4,9 ?
  • all the pictures in figs. 2, 4 and 9 would be better rotated by 90°; this would allow to better understand Figure 8, which reports the derivative along the laser line. Rotaing the pictures would also allow to explain that the laser line is orthogonal to the weld.

8- In line 140-142 the authors say that passing from RGB to grayscale reduces noise due to reflections.  I do not understand how this is possible, also because the laser is monochromatic. Can the authors explain ?

9-Lines 214-215-The authors should explain that the weld heigth is measured by a classical laser triangulation algorithm.

Author Response

We highly appreciate your comments and suggestions. Taking them into consideration we have made a revision of the paper, and the aspects pointed by you were addressed as follows:

  1. We have specified (for example, now in lines 20 and 293) that the post-welding defects detected by the proposed system are geometrical.
  2. We have deleted reference 1 since the term “pressure vessel” is well-known and it was not introduced in the deleted referenced. We have also deleted references 3 and 7.
  3. We have replaced “it is much more challenging to apply …” with “it is still a challenge to apply …” in line 34.
  4. We have included the missing term liquid in line 49.
  5. We have replaced the acronym TOFD with the term time-of-flight diffraction in line 49.
  6. We have replaced “3-D laser scanning systems” with “3-D Light Detection and Ranging (LiDAR) techniques” in line 54.
  7. We have redesigned Figure 1 to show a front and a side view of the system, and also a view where the fan angle of the laser is shown. We have included more information about the optical layout in the paragraph starting in line 113, stating in line 120 that the laser is orthogonal to the joint. We have also specified the angle between the optical axis of the camera and the laser, the fan angle of the laser and the stand-off distance of the laser triangulation system with respect to the inspected area in lines 116, 120-122 and in Figure 1. We have declared the size of the area captured by the camera in line 122, and specified that the horizontal size of the images in Figure 2 correspond to 65 mm. We have rotated 90º the images of Figures 2, 4 and 7 in order to facilitate the comprehension of Figure 6.
  8. We agree with you because the propose of passing to grayscale is just to keep only information about luminance as hue and saturation do not have relevant information for processing. We have rewritten lines 143-145.
  9. We have explicitly stated that the estimation of the weld bead dimensions is performed by a laser triangulation algorithm in lines 211-212.

Round 2

Reviewer 1 Report

  1. The title is not appropriate. "Computer Vision System for Welding Inspection of  Liquefied Petroleum Gas Pressure Vessels based on Combined Digital Image Processing and Deep 14 Learning techniques". Please reference.
  2. The description of the article needs to be improved. Such as "2.2 decision making"

Author Response

The authors highly appreciate the reviewer comments and suggestions. Taking them into consideration the paper was revised, and the aspects pointed by the reviewer were addressed as follows:

  1. The title was changed as the reviewer suggested.
  2. Sections 2.2 (lines 190-195) and 2.3 (lines 213-228) were partially rewritten in order to improve the description of the decision-making process.